# Biotic Factors Drive Woody Plant Species Diversity across a Relative Density Gradient of *Quercus aliena* var. *acuteserrata Maxim.* in the Warm–Temperate Natural Oak Forest, Central China

Chenyi Yu [1,†], Siyuan Ren [2,†], Yudie Huang [1], Guanjie Wang [1], Shengyun Liu [1], Zhenjiang Li [1], Yabo Yuan [1], Xin Huang [1] and Ting Wang [1,*]

[1] College of Forestry, Henan Agricultural University, Zhengzhou 450046, China; yuchenyi0914@163.com (C.Y.); 18338609018@163.com (Y.H.); drwhp1126@163.com (G.W.); lsy13225810180@163.com (S.L.); 18336457322@163.com (Z.L.); yuanyabo2216@outlook.com (Y.Y.); hxhanson@163.com (X.H.)

[2] China Aero Geophysical Survey & Remote Sensing Center for Natural Resources, Beijing 100083, China; rsy9999ml@163.com

[*] Correspondence: tingwang@henau.edu.cn

[†] These authors contributed equally to this work.

**Abstract:** Woody plants are crucial components of forest ecosystems and play critical roles in regulating community succession and ecosystem function. Studying woody plant diversity and its influencing factors is thus important for understanding and protecting forest ecosystems. *Quercus aliena* var. *acutiserrata* is an important deciduous broadleaf species in the warm–temperate forest of central China. Multiple regression and structural equation modelling were used to discuss the effect of biotic and soil factors on tree species diversity across seven relative density gradients of *Q. aliena* var. *acutiserrata* trees in this zone. Our results showed that the following: (1) Species diversity showed significant decreasing trends with increasing relative density of *Q. aliena* var. *acutiserrata*. (2) As the relative density of the oak tree increased, some biotic factors (canopy density, and mean DBH) and soil factors (Soil SOC, AP, and AK) all showed significantly increasing trends, whereas the DBH variation (CVD) and soil pH displayed decreasing trends. (3) Biotic factor (e.g., mean DBH, CVD, and competition interaction) had strong direct effect on species diversity, and soil factors exerted indirect roles on tree diversity via biotic factors. Our results provide insight into biodiversity protection and scientific management in this warm–temperate natural oak forest.

**Keywords:** species diversity; *Quercus aliena* var. *acuteserrata*; the Funiu Mountains; structural equation modelling; stand structural





## 1. Introduction

Species diversity refers to the richness and evenness of species distribution within a certain area, and it plays an important role in maintaining ecosystem function and stability [1]. Some potential factors affect species coexistence in forest communities, and several environmental factors, including biological interactions, climate, topography, and soil properties, can exert direct or indirect influence on species diversity [2]. It is particularly vital to explore the influence of environmental factors on species diversity in understanding forest community dynamics. Characterizing temporal changes in community composition and their influencing factors at a specific time point can offer insights into the underlying assembly processes [3]. However, our current understanding of how stage-specific factors influence species replacement during succession remains limited.

Forest community dynamics depend on stochastic processes in which temporal changes in community composition are affected by biotic and abiotic factors over time [4]. Biotic factors comprises numerous quantifiable biometric characteristics, such as stand

structure, competition interaction, and so on [5,6]. Stand structure refers to the spatial arrangement and attributes of the various components of the ecosystem, such as individual tree size variation among and within species, the size and height of the canopy, and the spacing of trees [7]. According to the previous theory, higher structural heterogeneity may accelerate the inequality between predominant and subdominant trees [8]. Furthermore, higher DBH variation often leads to more forest gaps, which result in more light and other resources for more individuals [9]. A previous study reported that by influencing light intensity within the forest, canopy density is seen as another important biotic factor in shaping the growth and distribution patterns of understory plants [10]. It was also concluded that canopy density can exert an indirect influence on species diversity within forest communities by producing an effect on environmental factors, such as light availability, temperature, and humidity, for understory individuals [11]. Another recent study also showed that asymmetric competition from multi-layered canopies can induce structural changes [12] and lead to variation in individual tree sizes (DBH variation) [9] within a community. The importance of competitive interactions gradually increases as resources become more limited along with the community succession process [13]. Consequently, the increasing competitive pressure will limit plant growth and succession, especially when asymmetric competition occurs [12]. In particular, smaller trees in the community are severely affected, resulting in reduced community stability and species diversity [14].

　　　Soil conditions (i.e., organic matter, available phosphorus, nitrogen, and available potassium, etc.), as abiotic environmental factors, affect the homogeneity of nutrients in the environment, resulting in a significant difference in biomass accumulation, species diversity, and ecosystem stability [15]. A previous study on dry deciduous forests in western India found that tree species richness was strongly positively correlated with soil nutrients, such as total nitrogen, available phosphorus, and organic matter concentrations [16]. However, a negative correlation between species diversity and soil nutrient content has also been shown in some studies. For example, Merunkova et al. found a significant negative correlation between alpine plant richness and soil organic matter content [17]. Huston et al. showed that species diversity decreased with increasing availability of soil resources (e.g., soil moisture and nutrients) [18]. It is possible that the correlation between plant diversity and soil resources when scaled up to community-level responses can be positive or negative, depending on the species assemblage and environmental influences during the different succession stages of the forest ecosystem [16,18]. Hence, it is crucial to clarify the effect of soil properties on plant diversity in forest ecosystems across the stages of stand development.

　　　A study on secondary succession from temperate forest revealed changes in species composition, stand height, and stem density during forest development. Notably, stem density, stand structure, and soil properties varied among different plant species types [19]. The temporal process of a dominant species plays a role in integrating community succession to some certain extent [20], and characteristics of such dominant species can be used to clarify the heterogeneity and potential dynamics within the process of forest succession. Moreover, the stem density of a dominant species can be considered as an important characteristic during its successional process [3,19], which involves dissimilar trees selecting adapted strategies in the understory and striving to reach the canopy through their growth process. In general, it is important to identify the key environmental factors affecting tree diversity along a dominant species' successional stages with different stem densities in order to improve our understanding of the mechanisms that determine community succession dynamics.

　　　The temporal process of a dominant species plays a role in integrating community succession to some certain extent [20], and longitudinal data on the dominant species within a community are needed to understand the mechanisms governing temporal changes in community composition. *Quercus aliena* var. *acuteserrata* is the dominant species in in the warm–temperate–subtropical transition zone of central China [21]. Several studies in oak natural forests have demonstrated that an increase in canopy density correlates with a decrease in plant species diversity [22,23]. Previous studies have mostly focused on species diversity and its influencing factors in subtropical forests, but few have compared the effects of biotic

and abiotic factors on species diversity in this zone. Based on previous studies, community-level attributes, specifically in terms of woody stem density [24], undergo transformations across various succession stages. Therefore, we selected different communities with distinct tree densities of *Quercus aliena* var. *acuteserrata* as a proxy for chronological process to discuss species diversity and its influencing factors along with the succession stages of the dominant species. We focused on changes in species composition of the oak community with seven relative density gradients (10%–70%) of the *Q. aliena* var. *acuteserrata* population. The main objectives of this study were to (1) describe species diversity and its biotic and abiotic factors at different successional stages of *Q. aliena* var. *acutiserrata*, and (2) compare the relative influences of biotic and abiotic factors on species diversity during the succession process of the dominant oak in the Funiu Mountains. The results serve as a reference for improving the general guidelines on community succession and provide a scientific basis for protecting and managing species diversity in mountainous communities.

## 2. Materials and Methods

### 2.1. Study Area

This study was conducted in the Baotianman National Nature Reserve ($33°25'-33°33'$ N, $111°53'-112°04'$ E) in the Funiu Mountains (Figure 1), which is located in the warm–temperate–subtropical transition zone of central China [21]. The study area is characterized by a subtropical monsoon climate with a mean annual temperature of 15.1 °C and mean annual precipitation of approximately of 886 mm. The main zonal vegetation types are categorized by dominant tree species, which include *Q. aliena* var. *acuteserrata*, *Quercus variabilis*, *Quercus serrata* var. *brevipetiolata*, *Pinus tabulaeformis*, and *Pinus armandii* [21], and high-altitude areas are dominated by *Q. aliena* var. *acuteserrata* trees. *Q. aliena* var. *Acutiserrata* is a heliophilous plant commonly found in acidic to slightly alkaline soil, and they prefer to grow in warm and humid environment [25]. The forest soil is classified as brown earth.

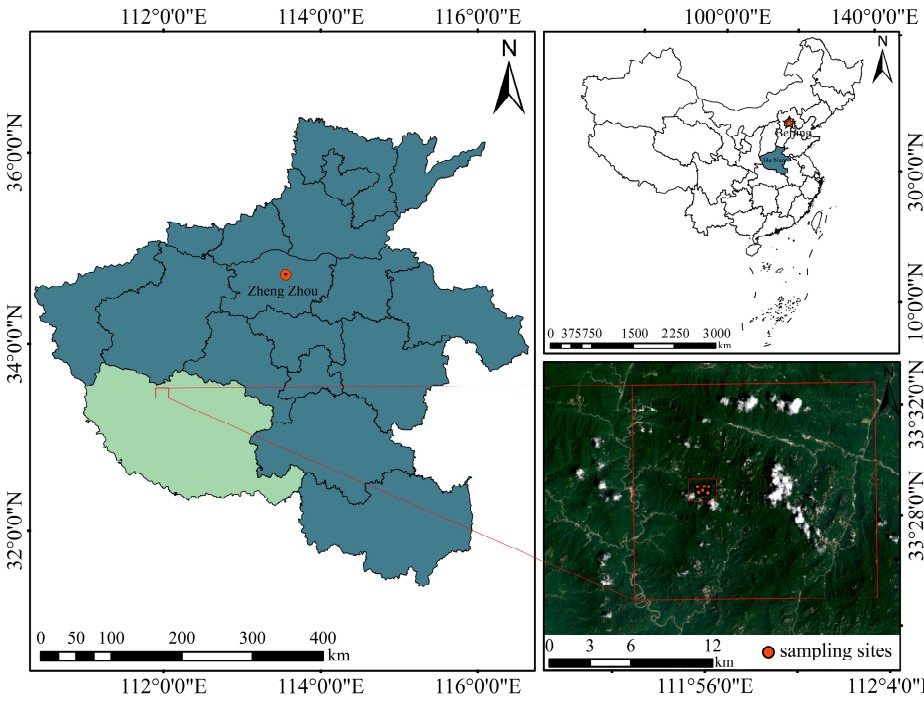

**Figure 1.** Location of the study site in the Baotianman National Nature Reserve, Henan Province, China.

### 2.2. Field Sampling and Vegetation Survey

In this study, we considered space instead of time by using the relative density of the *Q. aliena* var. *acuteserrata* population within the forest community instead of its succession stages. Seven relative density gradient (10%–70%) of *Q. aliena* var. *Acutiserrata* population were studied and three plots (20 m × 20 m) for each density gradient were selected in our

study. We classified the plots as monospecific or heterospecific and identified the dominant species or species group for each plot. These assessments were based on evaluating at least 80% of the live basal area of each plot.

Relative density refers to the percentage of the number of a species to the total number of all individuals in forest [25]. We selected different communities with distinct densities of *Q. aliena* var. *acuteserrata* individuals as the chronological process to discuss species diversity and its influencing factors along with the succession stages of the dominant oak species. Seven relative densities (10%–70%) of *Q. aliena* var. *acutiserrata* were selected, and three sample plots (20 m × 20 m) were set for each gradient. A total of 21 homogeneous rectangular plots (20 m × 20 m) with similar slopes were established in the Baotianman National Nature Reserve following the standard field protocol of the Center for Tropic Forest Sciences (CTFS, http://www.ctfs.si.edu, accessed on 18 June 2013). Basic information of the sampling plots were showed in Table 1. Species identity, height, DBH, localization, and crown width for all woody individual with a diameter at breast height (DBH) ≥ 1 cm were tagged, measured, and recorded. In each plot, individuals were enumerated into three DBH classes: trees (DBH ≥ 10 cm), saplings (10 cm > DBH ≥ 3 cm), and seedlings (1 cm ≤ DBH < 3 cm). All the recorded woody individuals (DBH ≥ 1 cm) in plots were identified by taxonomists and confirmed at https://www.cvh.ac.cn (Accessed on 1 August 2021).

**Table 1.** Basic vegetation characteristics of the established plots in the Funiu Mountain area in Henan Province, Central China.

| Relative Density (%) | Number of Species | Abundance | Sample Area (hm$^2$) | Mean DBH (cm) |
|---|---|---|---|---|
| 10 | 35 | 214 | 0.12 | 8.85 ± 0.76 a |
| 20 | 31 | 252 | 0.12 | 9.13 ± 1.59 a |
| 30 | 30 | 287 | 0.12 | 8.90 ± 1.29 a |
| 40 | 24 | 228 | 0.12 | 10.96 ± 1.44 a |
| 50 | 25 | 271 | 0.12 | 10.42 ± 0.26 a |
| 60 | 19 | 212 | 0.12 | 10.60 ± 1.10 a |
| 70 | 13 | 266 | 0.12 | 16.53 ± 2.89 b |

Notes: Different letters mean significant difference among different relative density ($p < 0.01$).

In each plot, soil samples were collected from five sampling points using a grid-based sampling method [26,27]. After removing dead surface leaves, small shrubs, and other herbs, the soil profile was excavated and soil samples were collected at three depths: 0–10, 10–20, and 20–30 cm. Samples from each of the three depth soil were then mixed together and stored in zip-locked bags for further analysis. Each soil sample was ~500 g, and a total of 315 soil samples were collected.

### 2.3. Soil Sampling and Chemical Analyses

Soil samples were transported to the laboratory for air drying, and then gridded to measure soil pH (pH), organic matter (SOM), total nitrogen (TN), available potassium (AK), and available phosphorus (AP). Soil TN was determined using the semi-micro Kjeldahl method, SOM was measured using the $K_2Cr_2O_7$ heating method [27]. Soil AK was determined using ammonium acetate extraction flame photometry, and soil AP was examined via alkaline hydrolysis NaHCO$_3$-extraction Mo–Sb–Vc colorimetry [17].

### 2.4. Diversity Index and Biotic Variables

In this study, the average diameter at breast height of all wood plants (DBH ≥ 1 cm) in each plot (mean DBH), the coefficient of variation of DBH (CVD), canopy density (CD), and the competition index (CI) in each plot were used as biotic factors [27,28] to investigate their effects on the species diversity of *Q. aliena* var. *acutiserrata* forest. The Simpson index (D), Shannon–Wiener index (N), and Pielou index (J) were selected to reflect the level of

community species diversity [27]. Based on the DBH and localization of every individual (DBH $\geq$ 1 cm) in each plot, CVD and CI were calculated by using the following formula.

$$CVD = \frac{\sum\limits_{i=1}^{n}(x_i - \bar{x})^2}{(n-1)\bar{x}} \tag{1}$$

where CVD is the coefficient of variation of DBH, $x_i$ is the DBH of the ith tree, $\bar{x}$ is the mean value of DBH, and n is the number of plants, and

$$CI = \sum\limits_{j=1}^{n}\left(\frac{D_j}{D_i}\right)\frac{1}{L_{ij}} \tag{2}$$

where CI is the competition index, $D_j$ is the competing wood DBH, $D_i$ is the DBH of the object wood, $L_{ij}$ is the distance between the object wood and competing wood, and n is the number of competing trees. The larger the CI value, the stronger the competitive pressure from the surrounding competing trees (i.e., competition is more intense).

*2.5. Statistical Analysis*

Unary linear regression was used to analyze the trends of biotic factors across its relative density and succession processes within the *Q. aliena* var. *acutiserrata* forest community. One-way analysis of variance (ANOVA) was used to examine significant differences in species diversity (Simpson index, Shannon–Wiener index, and Pielou index) along the stand proportions of the dominant *Q. aliena* var. *acutiserrata* trees. Linear regression was also employed to examine significant differences in soil nutrients and biotic factor along the relative density of this dominant oak trees. ANOVA and linear regression were statistically analyzed using SPSS 24.0 software (IBM, New York, NY, USA).

Multiple regression modelling (MRM) was used to examine the effects of multiple predictors on species diversity and its components. This analysis was conducted using the R package 'MuMIn' [29]. Structural equation modelling (SEM) was built to disentangle the direct and indirect drivers of species diversity based on a prior hypothesis of causal relationships that was informed by pre-existing knowledge of the mechanisms driving factor–species diversity relationships [1,13,16]. Maximum likelihood estimation was used to estimate path coefficients, and the fit statistics of the final model were evaluated using the comparative fit index (CFI), the goodness-of-fit index (GFI) and the standardized root mean square residual (SRMR). SEM analyses were performed using the R package 'lavaan' [9].

## 3. Results

*3.1. Associations of Relative Density with Tree Diversity*

The tree diversity of the oak communities with different relative densities of *Q. aliena* var. *acutiserrata* are shown in Table 2. The Simpson (D), Shannon–Wiener (N), and Pielou (J) indices reduced significantly with increasing relative density of *Q. aliena* var. *acutiserrata* ($p < 0.05$). The maximum index values (0.85, 2.34, and 0.83 for D, N, and J, respectively) achieved with the lowest relative density (10%) of *Q. aliena* var. *acutiserrata* stands, and the minimum index values occurred in plots with the highest proportion (70%).

*3.2. Associations of Relative Density with Biotic and Soil Factors*

Linear regressions revealed that all biotic variables varied across all forest plots with different proportions of *Q. aliena* var. *acutiserrata* trees (Figure 2a–d). As the relative density of *Q. aliena* var. *acutiserrata* in the stands increased, CD and mean DBH of the forest community increased significantly ($p < 0.01$), while the CVD for all individual trees decreased significantly ($p < 0.01$). CI showed a less obvious declining trend with increasing proportion of oak trees within these stands (Figure 2d, $p > 0.1$).

**Table 2.** Species diversity index of the oak community across a *Q. aliena* var. *acutiserrata* relative density gradient.

| Relative Density (%) | Simpson Index (D) | Shannon–Wiener Index (N) | Pielou Index (J) |
| --- | --- | --- | --- |
| 10 | 0.85 ± 0.07 a | 2.34 ± 0.29 a | 0.83 ± 0.13 a |
| 20 | 0.82 ± 0.07 ab | 2.14 ± 0.31 a | 0.80 ± 0.08 a |
| 30 | 0.82 ± 0.02 ab | 2.14 ± 0.10 a | 0.76 ± 0.04 a |
| 40 | 0.76 ± 0.01 bc | 2.22 ± 0.50 a | 0.72 ± 0.02 a |
| 50 | 0.72 ± 0.02 c | 1.91 ± 0.12 a | 0.69 ± 0.02 a |
| 60 | 0.62 ± 0.04 d | 1.33 ± 0.32 b | 0.69 ± 0.10 a |
| 70 | 0.45 ± 0.04 e | 0.98 ± 0.15 b | 0.50 ± 0.07 b |

Notes: Different letters mean significant difference among different relative density ($p < 0.05$).

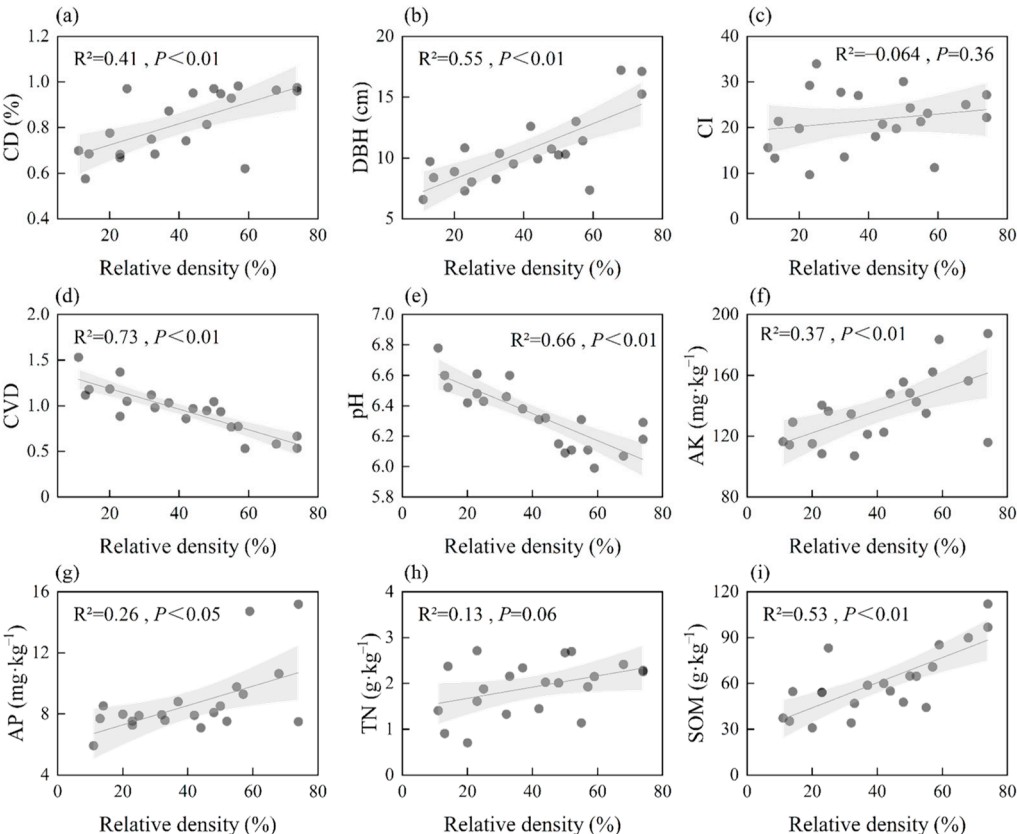

**Figure 2.** Biotic and abiotic factors of oak communities across a relative density gradient of *Q. aliena* var. *acutiserrata* trees. Filled areas indicate 95% confidence intervals. Abbreviations: (**a**) CD, mean canopy density; (**b**) DBH, mean average DBH of the community; (**c**) CI, mean competition index of *Q. aliena* var. *acutiserrata* community; (**d**) CVD, mean coefficient of variance of DBH; (**e**) pH, mean potential of hydrogen; (**f**) AK, mean available potassium; (**g**) AP, mean available phosphorus; (**h**) SOM, mean soil organic matter; (**i**) TN, mean total nitrogen.

Soil nutrients and its properties also varied with the proportion of *Q. aliena* var. *acutiserrata* trees in different forest communities (Figure 2e–i). Along with increasing proportion of *Q. aliena* var. *acutiserrata* trees in the stands, soil AP, AK, and SOM increased significantly ($p < 0.01$). In stands with a high proportion of *Q. aliena* var. *acutiserrata* (50%–70%), soil AP, AK, SOM, and TN reached their maximum values of 11.11 mg/kg, 160.30 mg/kg, 99.58 g/kg, and 2.46 g/kg, respectively. However, TN showed an unobvious upward tendency ($p = 0.06$), and soil pH displayed a significant downward trend ($p < 0.01$) along with increasing proportions of the dominant oak species.

### 3.3. Dominant Determinants for Tree Species Diversity

MRM was used to examine the combined effects of the factors driving species diversity of the oak communities along with a relative density gradient of *Q. aliena* var. *acutiserrata* trees (Figure 3). The mean DBH, CI, and CD were negatively related to the Simpson, Shannon–Wiener, and Pielou indices ($p < 0.05$) while CVD exerted a positive influence on the three diversity indices.

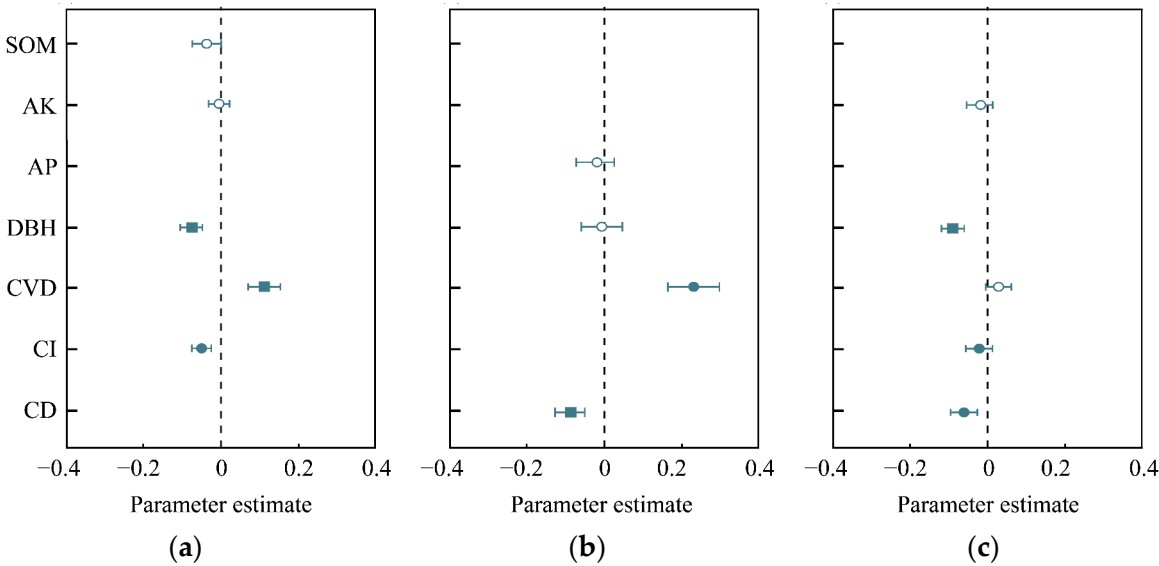

**Figure 3.** Effects of biotic and abiotic factors on tree species diversity in forest communities across a relative density gradient of *Q. aliena* var. *acutiserrata* trees. (**a**) Simpson index, (**b**) Shannon–Wiener index, (**c**) Pielou index. Notes: ○denotes no significancedifference at the 0.05 level. ● denotes significant difference at the 0.05 level; ■ denotes high significant difference at the 0.01 level. Abbreviations: CD, mean canopy density; CI mean competition index of *Q. aliena* var. *acutiserrata* community; CVD, mean coefficient of variance of DBH; DBH, mean average DBH of the community; SOM, mean soil organic matter; AK, mean available potassium; AP mean available phosphorus.

Soil nutrients (SOM, AK, and AK) also had a slight negative effect on the three diversity indices ($p > 0.05$). Specifically, CVD had a significant positive effect on the Simpson index, while DBH had the strongest negative effect on Simpson index, followed by CI (Figure 3a). CVD also exerted important positive effect whereas CD had a negative effect on the Shannon–Wiener index (Figure 3b). The Pielou index was significantly negatively effected by DBH, followed by CI in each plot (Figure 3c).

### 3.4. Direct and Indirect Effects on Tree Species Diversity

Biotic factors and soil factors were selected to construct SEMs (structural equation modelling) for discussing their direct and indirect effects on tree species diversity (Figure 4). The SEM results showed that CVD had a unique positive direct effect on the three diversity indices (0.63, 0.53, and 0.23 for D, N, and J, respectively). Both mean DBH and CI exerted significant negative and direct influences on the Simpson and Pielou diversity. Canopy density (CD) of forest communities also had an negative influence on the Shannon–Wiener diversity (−0.21). Soil factors (SOM, AK, and AP) indirectly affected Simpson, Shannon–Wiener, and Pielou diversity via biotic factors CVD and CI.

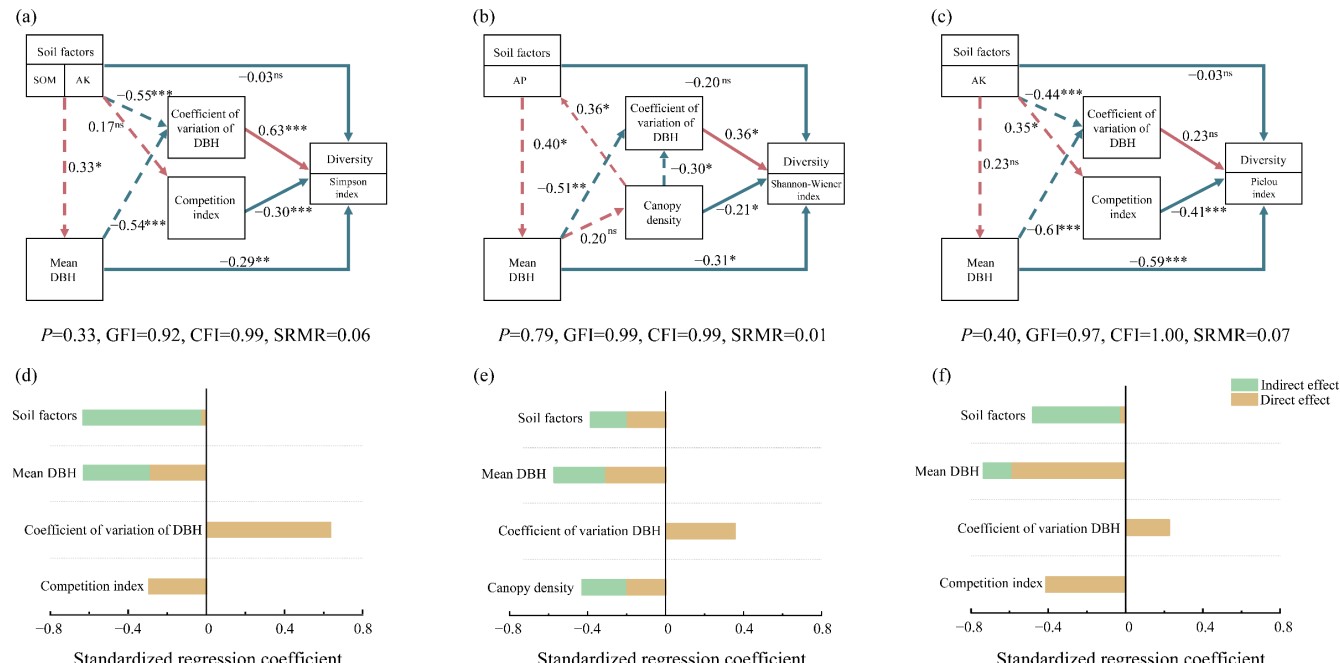

**Figure 4.** Optimal structural equation model (SEM) relating species diversity to predictor variables in the oak community across a stand density gradient of *Q. aliena* var. *acutiserrata* trees. Structural equation models (**a–c**) and standardized effects (**d–f**) in Simpson index disturbed quadrats (**a,d**), Shannon–Wiener index disturbed quadrats (**b,e**) and Pielou index disturbed quadrats (**c,f**). Numbers and associated asterisks are standardized path coefficients and significance levels (ns $p > 0.05$. * $p < 0.05$. ** $p < 0.01$. *** $p < 0.001$) for each path.

## 4. Discussion

### 4.1. Effects of Biotic Factors on Tree Species Diversity

According to GLMs and SEMs (Figures 3 and 4), tree species diversity in forest ecosystems is regulated by multiple biotic and abiotic factors. We found that DBH·variable (mean DBH and CVD) had a strong direct contribution to species diversity along a relative density gradient of *Q. aliena* var. *acutiserrata* (as the dominant species). Our result is similar to the findings of a previous study in temperate coniferous forests, in which individual tree size inequality (DBH variation) was found to be the dominant biotic driver of understory species diversity [9]. Our study is also consistent with two other research studies in subtropical oak forests [30,31], in which species diversity showed more correlation with biotic factors, such as stand density [30] and mean DBH [31]. The results of the present study demonstrated that CVD has the most positive effect, whereas mean DBH exerts a significant negative role on tree species diversity in these oak communities. It is theoretically plausible that higher DBH variation leads to large forest gaps, resulting in resources for more individuals in both horizontal and vertical spaces [9,32]. On the other hand, higher mean DBH from all individuals in each plot implies that similar large-diameter trees without a complementary structure cannot provide spatial resource allocation for a homogenous understory environment for more species. To some extent, this shows that the complex structure of the overstory has an important positive influence on species diversity in different *Q. aliena* var. *acutiserrata* communities.

*Q. aliena* var. *acutiserrata* is a deciduous broad-leaved plant that is the dominant species in warm–temperate natural forest [21]. As the relative density of *Q. aliena var. acutiserrata* trees increases, their large leaf blades may cause a gradual increase in the canopy density (Figure 2a). This oak species became the dominant species owing to its effective resource utilization rate, niche differentiation, and adaptation to environmental conditions in this zone. However, the gradual increase in canopy density of the *Q. aliena* var. *acutiserrata* community during succession may hinder more sub-story individuals from

obtaining matter exchange, resource capture, and water evaporation. Previous studies showed that the upper story canopy has a negative impact on the abundance of understory plants, which in turn leads to lower understory species richness [32]. Our previous study concluded that canopy gap disturbance can increase species richness stand density and alter species composition, which also suggests that a greater canopy gap attracts significant pioneer species supplementation with a favorable microenvironment [30]. Another study concentrating on mesic forest succession showed that 50% of species appeared in the early successional stage (in as early as the first 5 years of their regeneration dynamics) before perennial species generate a closed canopy [33]. As a result, the present study demonstrates that increasing stand density and canopy density may influence light and other resources, limiting species diversity within the successional process of *Q. aliena* var. *acutiserrata* community in this zone.

In addition, the present study is similar to the results of a previous research study, which demonstrated that larger densities of dominant species may result in asymmetric competition [34] and may be easier to capture for all types of resources [12], in turn resulting in higher dominance (Table 2 and Figure 2). Based on GLMs and SEMs, it is evident that competition interaction within the community significantly constrains species diversity. The advantages of the dominant species (i.e., higher stand density and canopy density) may infer a strong competitive advantage during successional development [12], while small and medium individuals are less able to access resources effectively owing to asymmetric competition. This also leads to a decline in small and medium individuals, which in turn results in a simplified DBH class structure. Nevertheless, a complex tree size structure is associated with improved light capture and use efficiency [35]. Thus, the competitive ability of a species determines its competitive exclusion owing to niche overlap, whereas species versatility determines niche differences between species [36]. This also exacerbates the overlaps and gaps between community ecological niches and reduces species diversity, resulting in the decline of other species.

### 4.2. Effects of Soil Factors on Species Diversity

There was a weak direct relationship between soil nutrients and species diversity (Figure 3a–f). Furthermore, our finding showed that soil factors, especially soil SOM, TK, and AP, indirectly influenced diversity indices by affecting some biotic factors, such as mean DBH, CVD, and CI. Similar results from previous studies also found that soil nutrients had indirect [32] and negative [37] impacts on aboveground biomass through species richness. In our study, most of the soil nutrients (soil SOM, and AK) had a negative direct effect on understory species diversity, which was mediated by their direct positive influences on DBH variation and CI of overstory individuals (Figure 4a,c). This, in turn, caused a greater direct effect on understory species diversity (Figure 4d,f). In addition, some negative influence from soil factors on species diversity could be mediated by their direct action on mean DBH, causing further indirect effects on DBH variation (soil SOM and AK, Figure 4a,c) and canopy density (soil AP, Figure 4b,e).

Plant species diversity varies with altitudinal and other gradients, which include composite factors, such as climate, soil characteristics, and biogeography [38]. In this study, the dominant species was *Q. aliena* var. *acutiserrata*, a deciduous broad-leaved tree, and its relative density gradually increased with the succession process. Our results show that stand density and nutrient accumulation in the soil increased as community succession progressed (Figure 3a–f). This is consistent with the findings of Klemmedson [39], who reported that oak trees density had an important association with understory production mediated by oak litter soil fertility. Consequently, the correlation between succession and canopy density showed a significant increase [21,40], implying that the amount of plant litter increases with community succession. Plant litter is an important source of organic carbon input in soil that can effectively improve the nutrient content of the soil [41]. With increasing soil nutrients, mean DBH and CD also increase, which consequently changes understory species composition. Our analysis suggests that soil nutrients can directly affect

the DBH of individual trees and CD, and indirectly affect species diversity through direct effects on light availability and plant structure.

## 5. Conclusions

Our study emphasizes the importance of biotic factors and quantifies the potential drivers on species diversity across a relative density gradient of dominant species, *Q. aliena* var. *acutiserrata*, in a warm temperate–subtropical transition zone forest. As the relative density of the oak tree increased, some biotic factors (canopy density and mean DBH) and soil factors (soil SOC, AP, and AK) all showed significantly increasing trends, whereas the DBH variation (CVD) and soil pH displayed decreasing trends. Biotic factors (e.g., mean DBH, CVD, and CI) were crucial for species diversity, and soil factors exerted indirect roles on tree diversity via biotic factors. The results give insights into the mechanisms that biotic and abiotic factors have the direct or indirect effects on tree diversity across the dominant oak tree's succession process. Moreover, the results could provide a theoretical basis for forest biodiversity conservation and the sustainable use of forest resources.

**Author Contributions:** Conceptualization, C.Y. and T.W.; methodology, S.R.; software, S.R.; formal analysis, Y.H. and G.W.; investigation, S.L., Z.L. and Y.Y.; writing—original draft preparation, C.Y.; writing—review and editing, T.W., X.H. and C.Y.; funding acquisition, T.W. All authors have read and agreed to the published version of the manuscript.

**Funding:** This research was supported by the National Natural Science Foundation of China (31270493) and the Pilot Project for Ecological Protection and Restoration of Mountains, Water, Forests, Fields, Lakes, and Grasses in South Taihang, Henan (JGZJ—Grant—2019125).

**Data Availability Statement:** Not applicable.

**Acknowledgments:** We would like to express our gratitude to Baotianman Forest Ecosystem Research Station.

**Conflicts of Interest:** The authors declare no conflict of interest.

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
