# Peer review of "Biotic Factors Drive Woody Plant Species Diversity across a Relative Density Gradient of Quercus aliena var. acuteserrata Maxim. in the Warm–Temperate Natural Oak Forest, Central China"

_forests, doi:10.3390/f14101956_

Round 1

Reviewer 1 Report

Comments and Suggestions for Authors

The subject is interesting (Biotic factors drive woody plant species diversity across a relative density gradient of Quercus aliena var. acuteserrata in the warm–temperate natural oak forest, central China), but the authors fail to write Ms clearly.  Particularly, the method section requires a lot of improvement, for example, how the stand density (SD), canopy density (CD), and the competition index (CI) were calculate. The introduction section needs lot of improvement, not clearly justify the title .  The first two paragraph in the introduction should be in context of the title. However, the data in the manuscript deserve publication and thus I give some advice with which the authors could add in the manuscript that could be published for example in this journal

Comments on the Quality of English Language

The subject is interesting (Biotic factors drive woody plant species diversity across a relative density gradient of Quercus aliena var. acuteserrata in the warm–temperate natural oak forest, central China), but the authors fail to write Ms clearly.  Particularly, the method section requires a lot of improvement, for example, how the stand density (SD), canopy density (CD), and the competition index (CI) were calculate. The introduction section needs lot of improvement, not clearly justify the title .  The first two paragraph in the introduction should be in context of the title. However, the data in the manuscript deserve publication and thus I give some advice with which the authors could add in the manuscript that could be published for example in this journal

Author Response

Dear reviewer:

Re: Manuscript ID: forests-2594461 and Title: Biotic factors drive woody plant species diversity across a relative density gradient of Quercus aliena var. acuteserrata in the warm–temperate natural oak forest, central China

Thank you very much for your recognition of our work and your valuable feedback. We have revised the paper according to your suggestions and comments, please find our revised part which has been marked in red. I have also carefully answered to all the annotations in your PDF document, and have organized each annotation in detail (Q1-Q16) and provided explanations. We would love to thank you and other reviewers for allowing us to resubmit a revised version of the manuscript and we highly appreciate your time and encouraging words .

Q1: Line 3. Write complete name. Quercus aliena var. acuteserrata Maxim.

Response 1: Thank you for the kind reminder. We have change “Quercus aliena var. acuteserrata” to “Quercus aliena var. acuteserrata Maxim.” in title. (Line 3, page 1)

Q2: Line16. Delete word modelling here.

Response 1: Thank you for the kind reminder. We have delete word modelling here. (Line 17, page 1)

Q3: The introduction section should start with importance of the forest structure then in second paragraph on species importance etc. I think starting introduction with species diversity didnot make any impact to readers.

Response 3: Thank you for your suggestion. We have rewritten the introduction according to your suggestion. (Line 30, page 1)

Q4:Introduction section is lengthy please reduce it.

Response 4: Thank you for the kind reminder. During the process of revising the manuscript, we also found that the introduction was too lengthy, so during the revision process, we minimized the part of the introduction as much as possible. (Line 30, page 1)

Q5: Figure 1. Add sampling sites in the map.

Response 5: Thank you for your suggestion. We have added the sampling points and legend in Figure 1 ,which makes the sampling location more intuitive.

Q6:Line 144. Please explain it in detail ...

Response 1: Thank you for the kind reminder. We have added detailed information on line 132-139, page 4.

Q7: Line 153. Shape of plot

Response 7: Thank you for the kind reminder. We have change “homogeneous square plots ” to “homogeneous rectangular plots” on line 162, page 4.

Q8: Line 155. It is quite confusing --- DBH of plant with grater than 10 cm are considered as tree.. then you count seedlings and saplings ... please clear it..

Response 8: Thank you for the kind reminder.The individuals we study include trees, seedlings and saplings. We have added detailed information to line 132.

Q9: Line 156. please provide details here..

Response 9: Thank you for the kind reminder. We have provided a detailed description of the soil sampling and analysis section.

Q9: line156. lmention number of soil sample

Response 9: Thank you for your suggestion. In our study the soil profile was excavated for a total of 105.Each soil sample was about 500 g and a total of 315 samples were collected. and added this number on line 159-161.

Q10:Line 158. explain you combine all samples together or analyzed each sample separate

Response 10: Thank you for your advise.In our study the soil sample were analyzed each separate.We have added a description in line162.

Q11:Line 170. no need of formula here...

Response 1: Thank you for your suggestion. We have removed the formula here.

Q12:Line 177. also mention whether you used seedlings and sapling data also

Response 12: Thank you for the kind reminder. The individuals we study include trees, seedlings and saplings. We have added detailed information to line 132,too.

Q13:Line 178. provide details of each parameter collected in the above section... how the canopy, stand density calculated.. also explain in detail about competition index .. what data you used to calculate.. also mention the parameters

Response 13: Thank you for the kind reminder. We have added formulas and explanations in the materials and methods on line186-189, page5.

Q14: Line 205. First explain what relative density means in methods section

Response 1: Thank you for the kind reminder. We have added a detailed introduction and formula on line 155, page4.

Q15:Line 215. What are the biotic factors ,, please explain it in methods also.. then readers can know.

Response 15: Thank you for the kind reminder. We have add what are the biotic factors is in line181-183.

Q16: line241. Mention what mean dBH is .. what was the class size.. values

Response 16: Thank you for the kind reminder. We have add the describe of mean DBH in line181.

Reviewer 2 Report

Comments and Suggestions for Authors

Dear authors!

The authors' article contributes to the understanding of the processes of succession of forest ecosystems in a separate territory. Determining the contribution of abiotic or biotic factors to the dynamics of the population structure of woody plants and the plant community as a whole will help to understand the productivity of natural ecosystems. The work has value in the applied aspect for the management of forest ecosystems.

However, I have a number of comments and recommendations for the authors.

1. Studies of the influence of the density of the location of trees in the forest on their taxonomic characteristics and the species composition of the plant community have been conducted for a long time. What is the novelty of your research?

2. L. 111, 135: the author's marks of plant species are not indicated.

3. In section 2, there is no indication of the systematic position of the model tree species, indication of its bioecological features – requirements for illumination mode, edaphic factors.

4. Section 2.2: There are no references to literature sources on methods of identification of plant species, determination of taxational indicators of trees, selection of soil samples.

5. Section 2.2: the volume of the material is not specified. How many individuals of trees have you taken the metrics? Further on, section 3 does not provide data on descriptive statistics of these indicators.

6. The authors used biodiversity indices, but do not provide information about the relative abundance of a species and the total number of species in a plant community.

7. Trees of different ontogenetic states grow in the plant community, which have different metrics. How did you take into account the factor of the biological age of individuals of the species in your study?

8. Also, the work does not touch on the discussion of the illumination factor as one of the key factors that determine the density of the plant population.

9. In section 2.6, you specify an ANOVA analysis, but further in section 3, the results of this analysis are not clearly spelled out – F and pairwise comparison of averages are not specified.

10. Table 1: There is no explanation of letter designations.

11. In section 4 there is no comparison of your own results with similar works on other tree species, especially on Quecus species.

12. Conclusions are not justified. There is no novelty in them.

Author Response

Dear reviewer:

Re: Manuscript ID: forests-2594461 and Title: Biotic factors drive woody plant species diversity across a relative density gradient of Quercus aliena var. acuteserrata in the warm–temperate natural oak forest, central China

Thank you very much for your valuable comments. We have revised the paper according to your suggestions and comments, please find our revised part which has been marked in red. During the revision process, as you commented, we found that our material methods were not rigorous and had errors. As the author of the article, I deeply apologize and have made serious revisions and proofreads. At the same time, we have carefully organized the conclusion of the article and pointed out our novelty regarding the unreasonable discussion and conclusion you raised.We would love to thank all reviewers for allowing us to resubmit a revised version of the manuscript and we highly appreciate your time and encouraging words.

Q1: Studies of the influence of the density of the location of trees in the forest on their taxonomic characteristics and the species composition of the plant community have been conducted for a long time. What is the novelty of your research?

Response 1: Previous studies have mostly focused on species diversity and its influencing factors in subtropical forests, but few have compared the effects of biotic and abiotic factors on species diversity in the warm temperate–subtropical transition zone. Based on previous studies, community-level attributes, specifically in terms of woody stem density , undergo transformations across various succession stages. Therefore, we selected different communities with distinct tree densities of Quercus aliena var. acuteserrata as a proxy for chronological process to discuss species diversity and its influencing factors along with the succession stages of the dominant species.

Q2: L. 111, 135: the author's marks of plant species are not indicated.

Response 2: Thank you for the kind reminder. We have changed the indicated species name in L. 106, 126.

Q3. In section 2, there is no indication of the systematic position of the model tree species, indication of its bioecological features – requirements for illumination mode, edaphic factors.

Response 3: We added the ecological characteristics of the dominant species,Q. aliena var. AcutiserrataQ. aliena var. Acutiserrata is a heliophilous plant with acidic to slightly alkaline soil, and they prefer to grow in warm and humid environment [26].”in line127 page 3

Q4. Section 2.2: There are no references to literature sources on methods of identification of plant species, determination of taxational indicators of trees, selection of soil samples.

Response 4: Thank you for the kind reminder. We have added relevant references for identification, indicator determination, and sample collection in the overall material method, and refined our soil sampling method to make the material method section of the article more scientific.

Q5. Section 2.2: the volume of the material is not specified. How many individuals of trees have you taken the metrics? Further on, section 3 does not provide data on descriptive statistics of these indicators.

Response 5:We have added a new Table 1, and added detailed numbers to the table.

Q6. The authors used biodiversity indices, but do not provide information about the relative abundance of a species and the total number of species in a plant community.

Response 6: Thank you for the kind reminder. We have adds the total number of species (S) in new table 1

Q7. Trees of different ontogenetic states grow in the plant community, which have different metrics. How did you take into account the factor of the biological age of individuals of the species in your study?.

Response 7: Thank you for your question. In the early stages of writing the article, we considered this issue and combined the density (table1), average diameter at breast height (Figure 2. b), and diameter class structure of the dominant species of Q. aliena var. Acutiserrata in the sample plot to determine the age structure of the Q. aliena var. Acutiserrata community.

Q8. Also, the work does not touch on the discussion of the illumination factor as one of the key factors that determine the density of the plant population.

Response 8: Thank you for the kind com. In this study, we did not quantify the lighting factor, mainly considering the relationship between canopies. The upper broad-leaved trees block sunlight, which leads to competition for lighting resources. Therefore, we compared other studies involving lighting in our discussion. In the revised discussion, we also deleted the discussion on lighting and replaced it with the capture of various resources.

Q9. In section 2.6, you specify an ANOVA analysis, but further in section 3, the results of this analysis are not clearly spelled out – F and pairwise comparison of averages are not specified.

Response 9: Thank you for your question. We have made changes after writing. Previously, we used ANOVA analysis to analyze all correlations, but later changed to only use ANOVA analysis for species diversity. However, our material and method sections were not modified in a timely manner. The correlation between variables and relative density was modified using univariate linear regression, which we have already made in the article (line.)

Q10: Table 1: There is no explanation of letter designations.

Response 10: Thank you for the kind reminder. explanation of letter designations have been add in line229,page6.

Q11. In section 4 there is no comparison of your own results with similar works on other tree species, especially on Quecus species.

Response 11:Thank you for your kind reminder. We have added the Previous research on subtropical oak forest in our discussion (line297,page9).

Q12. Conclusions are not justified. There is no novelty in them.

Response 12:Thank you for the kind reminder. We have rewritten our conclusion to make it more reasonable and highlight the novelty of our research.(line371,page10)

Reviewer 3 Report

Comments and Suggestions for Authors

line 110. The short description of the mountains shall go to next section, Material and methods.

line 52: instead of soil composition I would suggest soil conditions

line 296: „enough material exchange” could be better changed into matter exchange. material shall noun, not adjective.

line 335: latitudinal gradients. is the scope of the study so large to talk about latitudinal gradients. maybe altitudinal gradient is enough.

Author Response

Dear reviewer:

Re: Manuscript ID: forests-2594461 and Title: Biotic factors drive woody plant species diversity across a relative density gradient of Quercus aliena var. acuteserrata Maxim. in the warm–temperate natural oak forest, central China.

Thank you very much for your valuable comments. We have revised the paper according to your suggestions and comments, please find our revised part which has been marked in red. We would love to thank all reviewers for allowing us to resubmit a revised version of the manuscript and we highly appreciate your time and encouraging words .

Q1: line 110. The short description of the mountains shall go to next section, Material and methods.

Response 1: Thank you for your advices. We have incorporate the introduction to mountains into the materials and methods section of the introduction (line 120, page3).

Q2: line 52: instead of soil composition I would suggest soil conditions

Response 2: Thank you for your advices. We have revised the phrase throughout the manuscript .

Q3: line 296: “enough material exchange” could be better changed into matter exchange. material shall noun, not adjective.

Response 3: Thank you for your advices. We have change “enough material exchange” to “matter exchange” on line 318, page9.

Q4: line 335: latitudinal gradients. is the scope of the study so large to talk about latitudinal gradients. maybe altitudinal gradient is enough.

Response 4: Thank you for the kind reminder. After verification, We have found that our interpretation of this reference is indeed incorrect. So we have removed the latitude gradient and retained the altitude gradient on line 359, page 10.

Reviewer 4 Report

Comments and Suggestions for Authors

Thank you for your manuscript. I read through it and, by and large, the English is adequate but the paper still would benefit from review by a native English speaker. I understand that the particular combination of factors in your study is unique. I am not so certain that the methodology and practice are particularly novel, so I was hoping that your paper would have an interesting "twist" on the science. I did not find any such interesting hook, I'm afraid. 

I will recommend acceptance after revising the English. I found the methodology complete and useful as a reference.

Comments on the Quality of English Language

Scientfically sound I suppose, but not very interesting. It seems like the authors compared biodiversity to a basket of variables, just like one might compare the same basket of variables to productivity, or sunlight on the forest floor, or whatever.

I will recommend acceptance after revising the English. I found the methodology complete and useful as a reference.

Author Response

Dear reviewer:

Re: Manuscript ID: forests-2594461 and Title: Biotic factors drive woody plant species diversity across a relative density gradient of Quercus aliena var. acuteserrata Maxim. in the warm–temperate natural oak forest, central China

Thank you very much for your recognition of our work and your valuable feedback. We have made revisions according to the comments of all reviewers. Moreover the manuscript has been fully proofread by an English speaker, please find our revised part which has been marked in red. We would love to thank all reviewers for allowing us to resubmit a revised version of the manuscript and we highly appreciate your time and encouraging words.

Round 2

Reviewer 2 Report

Comments and Suggestions for Authors

Dear authors!
My comments were taken into account.